# Mango Peel Nanofiltration Concentrates to Enhance Anaerobic Digestion of Slurry from Piglets Fed with Laminaria

**DOI:** 10.3390/membranes13040371

**Published:** 2023-03-24

**Authors:** Antónia Macedo, Rita Fragoso, Inês Silva, Tânia Gomes, Cátia F. Martins, João Bengala Freire, Elizabeth Duarte

**Affiliations:** 1LEAF—Linking Landscape, Environment, Agriculture and Food, Associated Laboratory Terra, Instituto Superior de Agronomia (ISA), University of Lisbon, Tapada da Ajuda, 1349-017 Lisbon, Portugal; 2Polytechnic Institute of Beja, Campus do IPBeja, Rua Pedro Soares, Apartado 6158, 7801-908 Beja, Portugal

**Keywords:** mango peel, pig slurry, *Laminaria*, nanofiltration, sugar concentrates solutions, anaerobic digestion, anaerobic co-digestion, biogas production performance, biowaste valorisation

## Abstract

The environmental impact of biowaste generated during animal production can be mitigated by applying a circular economy model: recycling, reinventing the life cycle of biowaste, and developing it for a new use. The aim of this study was to evaluate the effect of adding sugar concentrate solutions obtained from the nanofiltration of fruit biowaste (mango peel) to slurry from piglets fed with diets incorporating macroalgae on biogas production performance. The nanofiltration of ultrafiltration permeates from aqueous extracts of mango peel was carried out using membranes with a molecular weight cut-off of 130 Da until a volume concentration factor of 2.0 was reached. A slurry resulting from piglets fed with an alternative diet with the incorporation of 10% *Laminaria* was used as a substrate. Three different trials were performed sequentially: (i) a control trial (AD_0_) with faeces resulting from a cereal and soybean-meal-based diet (S0); (ii) a trial with S_1_ (10% *L. digitata*) (AD_1_), and (iii) an AcoD trial to assess the effect of the addition of a co-substrate (20%) to S1 (80%). The trials were performed in a continuous-stirred tank reactor (CSTR) under mesophilic conditions (37.0 ± 0.4 °C), with a hydraulic retention time (HRT) of 13 days. The specific methane production (SMP) increased by 29% during the anaerobic co-digestion process. These results can support the design of alternative valorisation routes for these biowastes, contributing to sustainable development goals.

## 1. Introduction

Food production is facing the challenge of satisfying the demand of the increasing population. Higher production volumes lead to increased amounts of by-products/wastes, which need to be managed and valorised. In fact, the food waste hierarchy should be revised to reflect the new trends to be adopted, such as waste biorefineries in the circular bioeconomy [1]. Under the biorefinery concept, food waste is valorised in a “cascade”, extracting firstly potential value-added compounds and then using the extracted waste, for example, for energy recovery [2].

The intensification of livestock production, such as pigs, created environmental pressures and led to the development of alternative slurry management and valorisation practices. One possible valorisation route is the use of pig slurry as a substrate for bioenergy production through anaerobic digestion (AD) [3,4,5]. In this complex process, various groups of bacteria and substrates under anaerobic conditions can transform organic matter into biogas, a renewable fuel composed mainly of methane (CH_4_) and carbon dioxide (CO_2_) and other elements in smaller quantities, such as hydrogen (H_2_), nitrogen (N_2_), and hydrogen sulphide (H_2_S), and a high-quality organic-rich digestate [6,7]. AD has multiple environmental advantages, such as the production of renewable biogas, odour removal, the sanitisation of digestate, and a reduction in greenhouse gas (GHG) emissions [8]. However, the main drawbacks of the AD of pig slurry are the unfavourable C/N ratio due to the low concentration of carbon sources and the potential inhibitory effects of volatile fatty acids (VFAs) [9].

On the other hand, the search for alternative protein sources for animal feed promoted the development of studies on the dietary incorporation of microalgae, such as *Laminaria digitata*, a brown seaweed commonly found in North Atlantic shores and characterised by bioactive compounds such as laminarin. *L. digitata* has been shown to have beneficial effects on the gut health of weaned piglets and consequently on productive performance [10,11]. The fact that it is low in crude protein and fat may present an opportunity to use it, as it is high in the abovementioned polysaccharide and may have a prebiotic effect on the piglet digestive system. Laminarin’s ability to suppress pathogenic bacterial activity by lowering the adherence and colonisation of the mucosal surface has also been noted by authors as having antibacterial potential [12,13].

The weaning period is a stressful period for piglets due to all the alterations that occur, such as changes in social status, the environment, and nutrition. At a time when animals have underdeveloped digestive and immune systems, promoting gut health is crucial to mitigating post-weaning stress [14]. Thus, it was hypothesised that the use of *L. digitata* as an ingredient in the diets of piglets may produce a more favourable intestinal microbiota due to its prebiotic and antimicrobial potential, helping animals to go through this difficult stage more easily and reducing the utilisation of antibiotics and zinc oxide [15,16].

Simultaneously, the adoption of this alternative diet creates an opportunity to study changes in piglet faeces characteristics and the valorisation route, namely, AD performance. Another research trend regarding the use of AD for the valorisation of waste from livestock production is the addition of substrates that are richer in carbon sources, improving the C/N ratio and therefore the efficiency of AD [17]. In this case, the process is named anaerobic co-digestion (AcoD), because it involves more than one type of waste. AcoD allows some of AD’s mentioned drawbacks to be overcome by improving the nutrient balance and therefore microbial performance [18]. Moreover, it can be an innovative strategic approach for achieving a circular economy, closing the cycle of nutrients by producing organic fertilisers [8,9,19].

Fruit and vegetable wastes (FVWs) have physicochemical characteristics that make them appropriate for use as co-substrates, particularly given their high carbohydrate content [6]. This option overcomes possible problems with the management of this waste stream; for example, mango peel (representing about 15–20% of the fresh fruit) due to its carbohydrate, protein, and high water content [20,21,22,23], is susceptible to modification by microorganisms, creating environmental problems, such as the generation of leachates and gas emissions [24]. Previously research concluded that the utilisation of mango peel showed good potential for biogas production, with low contents of CO_2_ and H_2_S [25,26]. Therefore, it is likely that the addition of concentrated sugar solutions obtained from mango peels, which are rich in carbon sources, can contribute to improving the C/N ratio during the AcoD process with pig slurry and the efficiency of biogas production.

Membrane separation processes have been used to recover high-value compounds from FVWs because they present several advantages over other processes; for example, (i) they can be performed at room temperature, with reduced energy consumption, and at the same time, they protect food wastes from thermal degradation; (ii) they are easy to implement due to their modular character; and (iii) several membrane materials and membranes with different porosities are available on the market [27,28,29]. In a previous work, sugars from mango peels were recovered through the following sequence of processes: the extraction of soluble compounds with water and filtration through cotton cloths to remove suspended solids, followed by a sequence of membrane processes, such as ultrafiltration (UF), with polysulphone membranes with a cut-off of 25 kDa, and nanofiltration (NF) of UF permeates, using polypiperazine membranes with a cut-off of 130 Da. Higher-molecular-weight carbohydrates were completely removed by the ultrafiltration process, whereas those with lower molecular weights were transferred to the permeates. The concentration of ultrafiltration permeates by NF to a volume concentration factor of 2.0 led to the removal of more than 90% of lower-molecular-weight and fermentable sugars, such as glucose, fructose, galactose, sucrose, and maltose [23].

This work aims to fill the knowledge gap on the effect of NF concentrates from mango peel (co-substrate) incorporation on the AD of pig slurry (substrate), as, to the best of the authors’ knowledge, it has not been previously addressed in the literature. The presented results, obtained from a laboratory-scale AcoD study to evaluate the potential for bioenergy, provide insight into this innovative biowaste management strategy.

## 2. Materials and Methods

### 2.1. Sample Collection, Pre-Treatments, and Feed Mixtures

#### 2.1.1. Substrates

Piglet faeces were collected from an experimental trial using twenty post-weaning male piglets from Large White sows crossed with Duroc boars with an initial live weight of 10.5 ± 0.62 kg. It should be mentioned that following the principles of European Union legislation (2010/63/EU Directive), the procedures used in animal experimentation were revised by the Ethics Commission of Instituto Superior de Agronomia and accepted by the National Authority for Animal Health (Process number 0421/2020; Direção Geral da Alimentação e Veterinária: Lisboa, Portugal).

Each animal was allocated to a metabolic cage equipped with a feeder, a stainless-steel nipple, a heating lamp, and plates for the separation of faeces and urine, as described elsewhere [14]. The animals had 5 days for environmental and diet adaptation and the stabilisation of their stress and digestive conditions, followed by an experimental period of 14 days. Each animal had access to one of the experimental diets: (1) a cereal- and soybean-meal-based diet (control, n = 10); (2) a control diet with 10% *L. digitata* (LD, n = 10), as can be seen in Figure 1. *L. digitata* used in the experimental diets was provided as dried powder by the company Aleor (Lézardrieux, France).

Piglets were fed daily, with an equal amount per animal (645 g/day/animal), and had free access to water. Feed refusals were collected and weighed daily. Total excreta, faeces, and urine were collected separately during the entire experimental period. Fresh faeces from the two experimental diets were then homogenised and mixed to be used as substrates for anaerobic digestion and co-digestion trials. Faeces from the cereal- and soybean-meal-based diet, designated as A, were the control for AD trials (AD_0_), and faeces from the diet with 10% *L. digitata* incorporation, designated as B, were used for AD_1_ trials (Figure 1).

Pre-treatments were decided based on the sample’s characteristics (presented in Section 3, Table 1). This procedure (Figure 1) included hydrolysis (100 g of fresh faeces in 800 mL of water), followed by mechanical stirring (using a VELP Scientifica, Usmate, Italy) for 2 min at 50 rpm, 35 W, to homogenise the mixture.

#### 2.1.2. Co-Substrate

The preparation of NF concentrates for AcoD trials followed the scheme presented in Figure 2. Since a detailed description of their preparation was already described elsewhere [23], in this study, only the main procedures are summarised. Mango peels from different varieties were supplied by a local market in Beja, Portugal, and characterised. After being received at the lab, the samples underwent a cold-water wash, superficial drying with a cotton cloth, and size reduction by milling. The extraction of the soluble compounds present in mango peels was performed with hot water, at a temperature of 70 °C, using a solid-to-liquid ratio of 1:10.

The aqueous extracts were subjected to UF until a volume concentration factor (VCF) of 2.0 was reached. The UF membranes used had a cut-off of 25 kDa and a surface-active layer of polysulphone, and the membrane area used was 0.072 m^2^. The best experimental conditions for this membrane/feed were previously studied through experiments in total recirculation mode. So, the concentration of the aqueous extracts was carried out at a transmembrane pressure of 2.0 bar, at a feed circulation velocity of 0.91 ms^−1^, and at room temperature.

The UF permeates underwent nanofiltration until a VCF = 2.0 was reached using NF membranes with a cut-off of 130 Da, a surface-active layer of polypiperazine, and a total surface area of 0.072 m^2^. The best operating conditions selected to produce NF concentrates, after carrying out total recirculation experiments, were a transmembrane pressure of 20.0 bar and a feed circulation velocity of 0.91 ms^−1^, and they were kept at room temperature [23]. The physicochemical characterisation of NF concentrates, designated as S_2_ in Figure 2, is shown in Table 2.

The same Lab-Unit M20 module from Alfa Laval, Denmark, was used for all the permeation studies (Figure 2). A maximum of 20 pairs of flat-sheet membranes, each having an area of 0.018 m^2^, may be employed in this plate and frame module.

#### 2.1.3. Feed Mixture

For AcoD trials, S_1_ was the chosen substrate according to its pH values, which were more suitable for the anaerobic digestion process. To valorise the by-product resulting from the membrane technology, a co-digestion trial was carried out using NF concentrates from mango peel as a co-substrate (S_2_). AcoD trial feed consisted of a mixture of S_1_ and S_2_ in an 80:20 (%) ratio. This mixture was then stirred (VELP Scientifica, 50 rpm, 35 W) for 1 min to make sure it was homogenised (Figure 3).

### 2.2. Experimental Design and Operational Conditions

Anaerobic digestion and co-digestion experiments were performed in a continuous-stirred tank reactor (CSTR), which had a 4.8 L working volume, under mesophilic conditions (37.0 ± 0.4 °C), with a hydraulic retention time (HRT) of 13 days for each trial. The CSTR was controlled by computer software, which monitors and controls the mechanical stirrer (VELP Scientifica, Italy, 50 rpm, 60 W) and the heating system. The software also controls the feeding pump (Watson Marlow, UK, 120 rpm) through which the reactor was fed.

The biogas produced during AD/AcoD was measured with a flow meter (µFlow, bioprocess Control, Germany), and the digestate was also characterised.

Three different trials were performed sequentially: (i) a control trial (AD_0_) with faeces resulting from a cereal- and soybean-meal-based diet (S_0_); (ii) a trial with S_1_ (10% *L. digitata*) (AD_1_), and (iii) an AcoD trial to assess the effect of the addition of the co-substrate (20%) to S_1_ (80%).

### 2.3. Performance and Stability Operational Parameters

During AD and AcoD experiments, the HRT was kept at 13 days, with the OLR (organic loading rate) ranging between 1.32 ± 0.16 g VS/L_reactor_.d and 2.09 ± 0.13 g VS/L_reactor_.d.

The following parameters were monitored during experiments: pH, electrical conductivity (EC), biogas production, and biogas quality, which were measured through an analytical device connected to the reactor’s biogas flux bypass (LMSxi Multifunction Landfill Gas Analyser, Gas Data, Coventry, UK). Other performance parameters were calculated, such as gas and methane production rates (GPR and MPR, respectively), specific gas and methane production (SGP and SMP, respectively), and the specific energy loading rate (SELR). The SMP was selected as an indicator to allow a comparison among the trials performed and easily evaluate the energy recovery potential for the overall AD system. The SELR is an indicator that can be used to measure microbial activity, and under normal circumstances, it should be less than 0.4 d^−1^, indicating that the process is stable. There is a correlation between the daily organic feeding load (expressed in terms of TCOD) and the reactor’s biomass through the volatile suspended solids (VSS) content [30].

### 2.4. Analytical Characterisations

The characterisations of samples, substrates, feed mixtures, and digestates were performed according to the American Public Health Association (APHA) [31] and included the determination of pH and EC (both determined using Multi 3430, WTW, Germany), total solids (TS), volatile solids (VS), VSS, total chemical oxygen demand (TCOD), ammoniacal nitrogen (N-NH_4_^+^), and Kjeldahl nitrogen (TKN). Total organic carbon (C_ORG_) was determined according to previous authors [32], and the carbon/nitrogen ratio was calculated by calculating the ratio between C_ORG_ and TKN.

Concentrates from nanofiltration experiments, S_2_, were analysed for pH, using the potentiometric method with a potentiometer (Methrom 744 pH Meter; METHROM LTD. company, Herisau, Switzerland); TS, VS, and TCOD, in accordance with APHA [31]; TKN, using the *Official Methods of Analysis of AOAC* [33]; and the most fermentable sugars (glucose, galactose, fructose, and sucrose), using High-Performance Liquid Chromatography/Ion Chromatography (HPLC/IC) [34] with a CarboPac PA10 column (Dionex, Sunnyvale, CA, USA) equipped with an amperometric detector and the procedure described in [33]. The analysis was performed at 30 °C with sodium hydroxide (NaOH 4 mmol/L) as the eluent at a flow rate of 0.9 mL/min, and standards of glucose, galactose, fructose, and sucrose (Panreac Quimica SAU, Barcelona, Spain) in the range of 0.006–0.2 g/L were used.

### 2.5. Statistical Analysis

All analyses were performed in two or three replicates, and the results are shown together with the means and standard deviations for each.

Statistical analysis was conducted in GraphPad Prism Software (version 5.0). Analysis of variance (ANOVA) was performed using the Tukey test to compare three or more samples and Student’s t-test to compare two samples, both with a 95% degree of confidence (*p* = 0.05). When *p* values were less than 0.05, differences were considered significant.

## 3. Results and Discussion

The findings are supported by a reasoning strategy wherein substrates S_0_ and S_1_ (which came from piglets fed a control diet and those fed a diet with 10% *Laminaria* introduced, respectively) were first assessed. After investigating the impact of including *Laminaria digitata* in AD, AcoD trials were conducted utilising S_1_ as the primary substrate (which showed better results in AD trials) and with the addition of nanofiltration concentrates (S_2_) obtained from mango peel as a co-substrate. AD and AcoD experiments were evaluated in terms of process performance and stability.

### 3.1. AD Trials

Following the schematic origin of the different faeces samples (A and B), their physicochemical characteristics are reported in Table 1.

Regarding the TS values, it is possible to state that sample B has a lower value than sample A (a decrease of around 5%). However, the nitrogen content of the B sample was slightly lower than that of the A sample, showing an improvement of 7% in the C/N ratio, and, consequently, can be beneficial for the fermentation process.

These findings are consistent with some authors’ [35] reports of C/N ratio values for post-weaning piglet slurry in the range of 14.5, as well as with other authors’ [36] usage of pig dung and presentation of a C/N ratio value of 13.45. To make samples A and B viable for bioconversion in a wet AD system, it was necessary to adopt some pre-treatment procedures. The objectives for the selection of the pre-treatment were to obtain substrates (S_0_ and S_1_) with similar physicochemical characteristics to pig slurry, mainly the OLR, to allow a comparison with previous studies [37,38].

Table 3 shows the physicochemical characteristics of the feeds and digestates obtained in AD trials.

The results in Table 3 highlight the positive impact on the pH of the AD feed due to the incorporation of *Laminaria digitata*. This increment is favourable for the bioconversion process, in accordance with the values recommended for AD (6.5–7.5) [39].

The 15% rise in the C/N ratio that resulted from the addition of *Laminaria digitata* to the feed is another important factor in improving the AD process. This value is closer to the range of values that are ideal for the AD process, which is between 15 and 30 [40]. It is important to achieve a well-balanced C/N ratio since methane production will be low due to limited carbon and increasing ammonia concentration if the C/N ratio is unbalanced [41]. Comparing C/N ratios from this experiment with a study carried out with pig slurry from the fattening/finishing phase [38], AD_0_ corresponded to an increase of 86%, and AD_1_ doubled this parameter.

The TS and VS values are in line with those reported by previous authors [42] (34.6 g/kg and 24.5 g/kg, respectively), who used pig manure as the main substrate. The same applies to the TCOD value, which, in the same study, was 33.7 g/L.

The statistical analysis of AD_0_ and AD_1_ characteristics confirmed that there were no significant differences (besides pH and EC) between trials (*p* > 0.05), meaning that faeces from conventional diets can be replaced by faeces from piglets fed diets fortified with 10% *L. digitata*.

Regarding the TCOD and VS removal efficiency, the performance of the two mono-digestion runs was similar, showing values of 67–68% and 68–71%, respectively. In both trials, the pH of the digestate indicated the AD process’ stability.

During AD trials, the definition of key operational parameters is crucial to evaluating the performance and stability throughout the AD_0_ and AD_1_ assays. Table 4 summarises the values obtained to allow a comparison between AD_0_ and AD_1_.

As shown in Table 4, the OLR values did not have significant differences during the two trials. For other operational parameters, such as the GPR and MPR, the values presented also do not have significant differences, so the process performance was not affected by the incorporation of 10% *Laminaria digitata* into the feed, as supported by the statistical analysis (*p* > 0.05). The SELR value is below 0.4 d^−1^ for both trials, indicating that the ratio between the organic load and the reactor’s biomass is appropriate for a stable bioconversion process [30].

### 3.2. AcoD Trials

Given the improvement of the feed characteristics due to the introduction of *L. digitata*, co-digestion trials were performed using feed S_1_ as a substrate.

The co-substrate was defined based on a previous work [23], where it was concluded that NF concentrates from mango peel could be utilised as a co-substrate, so the characterisation of this concentrate was carried out to assess its incorporation in AcoD trials. Table 2 presents the characteristics of S_2_, which facilitate the process of selecting the proportion of the co-substrate suitable to incorporate with the selected substrate of AD.

In Table 2, it can be observed that the organic matter present in co-substrate S_2_ (nanofiltration concentrate) is mainly composed of monosaccharides and disaccharides, comparing their concentrations with that of volatile solids, VS, and with results obtained elsewhere [23]. The concentrations of TKN were lower because most of the organic nitrogen was retained by the previous ultrafiltration process. Using a similar strategy to that utilised in this study, other researchers [43] who attempted to recover polyphenols from spinach and orange by-products came to the same conclusion that the polyphenols could be separated from simple sugars using NF membranes with a cut-off of 300 Da. In this case, the polyphenols were retained by NF membranes, while simple sugars were recovered in permeates [43]. In this research, the use of NF membranes with a much lower cut-off (130 Da) allowed the retention of the most fermentable sugars in the concentrates, as intended for the anaerobic co-digestion process. In fact, the C/N ratio obtained for co-substrate S_2_ was much higher than that obtained for S_1_ (Table 3).

This factor guarantees that S_2_ will be an adequate co-substrate for S1, as it will aid in increasing the C/N ratio of the feed mixture to a ratio that is between 15 and 30, which is more favourable for the anaerobic digestion process [40].

The use of 20% integration of S_2_ was chosen to provide a balanced synergetic effect and to follow recommendations for the AcoD process, since the pH of S_2_ is lower than the process’ optimal value, and, in contrast, the high C/N ratio value (56) could endanger the feed mixture.

To support the above statements, a ratio of 80:20 S_1_:S_2_ is suitable to maintain the performance and stability of the AcoD assay. With this choice, the feed combination shown in Table 5 displays pH and C/N ratio values that are within the range advised by other researchers [39,40]. Table 5 presents the feed mixture’s physicochemical characteristics, as well as those of the digestate, after the AcoD process.

The most relevant change resulting from the introduction of S_2_ as a co-substrate was a 20% increase in C/N, as expected, because NF concentrates from mango peel mainly contributed carbon (as seen in Table 2). S_2_ resulted from a sequence of membrane processes (UF-NF), where most of the organic matter was retained by UF membranes. As a result, NF concentrates have a reduced solids content, which lowers feed VS by about 17%. The C/N ratio reached in AcoD experiments increased by 80% when compared to prior research employing pig slurry and pineapple peel waste, with an OLR of 1.45 ± 0.02 g VS/L_reactor_.d [38].

Table 6 summarises the performance and stability parameters for AcoD trials, which allowed the evaluation of the effect of the incorporation of the 20% NF as a co-substrate.

It is important to mention that the OLR of AcoD is 34% lower than that of AD_1_ (2 g VS/L_reactor_.d, Table 4).

It was feasible to assess the process’ stability by comparing the SELR values between AD_1_ (0.38 ± 0.03 d^−1^) and AcoD; the biogas quality exhibited the same behaviour (expressed as a percentage of CH_4_). These results confirm that the process was not compromised by the change to AcoD. To determine the effect of the incorporation of both *L. digitata* and the co-substrate, a statistical analysis was performed on the pH values of the feed and digestate. To represent this analysis, a box-and-whisker plot was created (Figure 4) with a five-number summary: the minimum, the 25th percentile, the median, the 75th percentile, and the maximum, with the whiskers extending to the minimum and maximum.

In Figure 4i, it is possible to observe the positive effect of *L. digitata* incorporation on the feed pH. The control trial (AD_0_) had pH values of 5.4 ± 0.2, and, with macroalgae incorporation, these values had an increase of 20%, reaching 6.5 ± 0.3. Based on the statistical analysis performed, it is possible to conclude that there was a significant increase (*p* < 0.05). This fact is relevant because pH values should be in an ideal range for anaerobic digestion to occur steadily and increase biogas generation (6.5–7.5). Below 6.5, the methanogenesis growth rate could be reduced, and for values higher than 7.5, a system failure can occur [39].

For AcoD, although the co-substrate addition had a positive effect on pH values (6.7 ± 0.3), this difference was not statistically significant (*p* > 0.05).

Regarding the pH values of the digestate from AD trials (Figure 4ii), it is possible to observe that there were no major fluctuations: the pH in the trials remained near neutrality [38]. However, when statistically analysing the values, in contrast to the feed pH, there were no significant differences between AD_0_ and AD_1_ (*p* > 0.05).

As mentioned before, the SMP parameter was suitable to compare the performance of the different trials to offset the variation in the OLR. Figure 5 illustrates the average SMP values for each trial.

During the AD_1_ trial, an increase of about 8% in methane yield occurred in comparison with the AD_0_ assay. Values from other studies are in line with those achieved in this experiment: one of them reports an SMP of 220 mL/g VS for AD trials performed with pig faeces with a control diet [41]; another one, also for AD with pig slurry, indicates an SMP of 212 mL/g VS [39].

The adoption of a co-digestion regime led to a more relevant increase in SMP of 29%, indicating that bioconversion was more efficient. This fact is probably associated with the more balanced C/N ratio caused by the co-substrate addition. This statement is confirmed by other authors who noted that combining co-substrates with animal manure improved the C/N ratio, which led to an increase in methane yields [8,39]. For example, the co-digestion of pig slurry with pear waste led to an SMP of 243 mL/gVS, which corresponded to an increase of 35% compared to pig slurry mono-digestion [4].

## 4. Conclusions

Concerning AD assays, it is important to highlight the positive effect of L. *digitata* incorporation on the feed pH. The feed in the control trial (AD_0_) had pH values of 5.4 ± 0.2, and with macroalgae incorporation, these values increased by 20%, reaching 6.5 ± 0.3, achieving the range recommended for anaerobic biological process.

Regarding the AcoD trials, combining slurry from piglets fed with 10% *Laminaria* and NF concentrates from mango peel biowaste proved to be a suitable solution for increasing the amount of energy generated. The experimental tests highlighted synergic effects, which, in the given operating conditions, led to higher methane production than that obtainable by processing pig slurry as a mono-substrate. Even with a slight decrease in the OLR, the results obtained showed quite stable operating conditions, without inhibition phenomena. The integrated pre-treatments of piglets’ faeces and mango peel also contributed to higher AD and AcoD performance, as organic matter availability increased during the process.

The environmental benefits achieved with the proposed strategy include the recovery of renewable energy, replacing fossil fuels and therefore avoiding associated greenhouse gas emissions. Another advantage is the production of an organic fertiliser that can restore soil organic carbon and be used as a substitute for the use of mineral fertilisers and, consequently, reduce the carbon footprint associated with their production.

The integrated management of fruit biowaste and piglet slurry by co-digestion can contribute to achieving bioeconomy targets and sustainable development goals, promoting the sustainability of food production.

Therefore, further studies should be developed with the aim of optimising the proposed solution. For example, regarding the co-substrate, it would be interesting to prepare NF concentrates from other fruit wastes or from a mixture and to optimise the pre-treatment to avoid unnecessary dilution with water. The introduction of higher percentages of the co-substrate should also be tested.

## Figures and Tables

**Figure 1 membranes-13-00371-f001:**
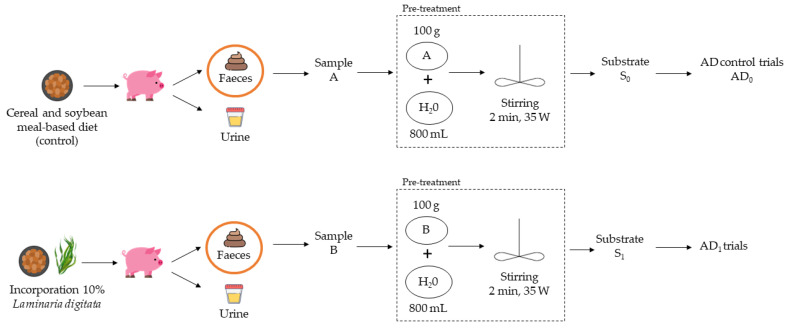
Substrate origins and pre-treatments for AD trials. Experimental conditions: mixing at 50 rpm, 35 W, for 2 min at room temperature.

**Figure 2 membranes-13-00371-f002:**
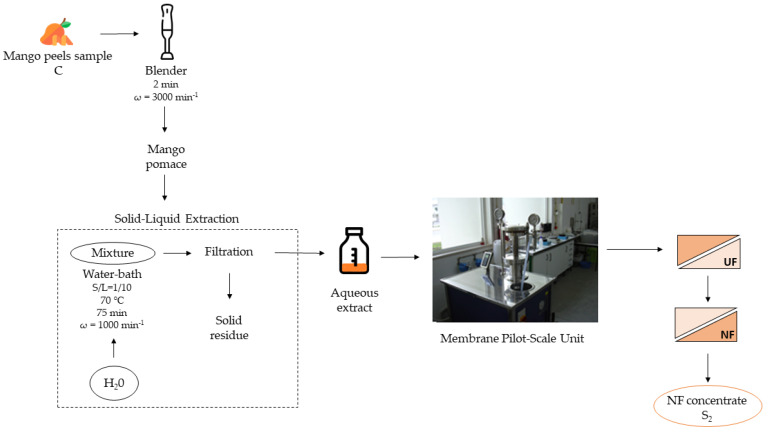
Co-substrate origin (adapted from [23]). Experimental conditions—UF: membranes with a cut-off of 25 kDa; ΔP = 2.0 bar; v = 0.91 m s^−1^, at room temperature. NF: membranes with a cut-off of 130 Da; ΔP = 20.0 bar; v = 0.91 m s^−1^; at room temperature.

**Figure 3 membranes-13-00371-f003:**
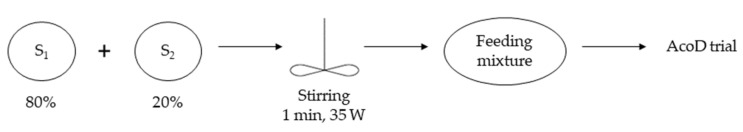
Feed mixture preparation for AcoD trials. Experimental conditions: mixing at 50 rpm, 35 W, for 1 min at room temperature.

**Figure 4 membranes-13-00371-f004:**
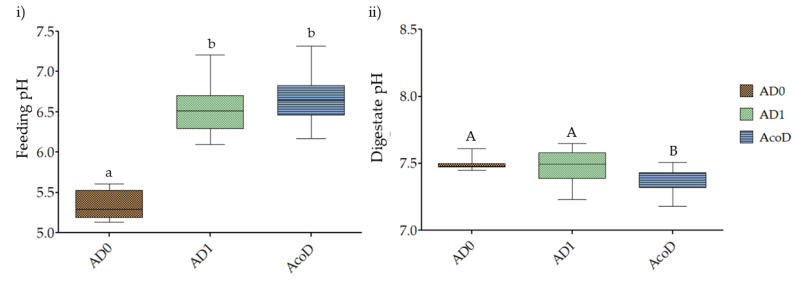
(**i**) Feed and (**ii**) digestate pH evolution throughout AD and AcoD trials. Different letters in the graphic indicate significantly different results (*p* < 0.05) according to the Tukey test.

**Figure 5 membranes-13-00371-f005:**
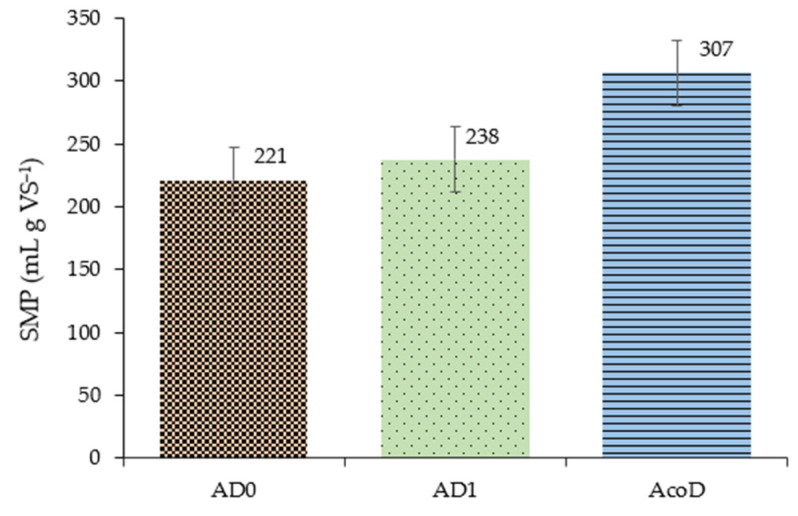
SMP evolution through AD/AcoD trials.

**Table 1 membranes-13-00371-t001:** Characterisation of samples A and B.

	A	B
TS (g/kg)	291 ± 0.46	276 ± 0.13
VS (g/kg)	257 ± 0.39	243 ± 0.12
VS/TS (%)	88	88
TKN (g/kg)	10.7 ± 0.12	9.7 ± 0.04
C_ORG_ (g/kg)	149	141
C/N	14	15

**Table 2 membranes-13-00371-t002:** Characterisation of co-substrate S_2_.

	S_2_
TS (g/L)	11.0 ± 0.5
VS (g/L)	2.9 ± 0.05
Galactose (mg/L)	<1
Glucose (mg/L)	116.9 ± 5.1
Fructose (mg/L)	1958 ± 30.2
Sucrose (mg/L)	856.9 ± 43.2
TKN (g/L)	0.03 ± 0.004
C_ORG_ ^1^ (g/L)	1.68
C/N	56

^1^ C_ORG_ was calculated theoretically based on the concentrations of the most fermentable sugars.

**Table 3 membranes-13-00371-t003:** Feed and digestate characterisation for AD trials. Different letters in the table indicate significantly different results between AD_0_ and AD_1_ (*p* < 0.05) according to Student’s *t*-test. Parameters were not compared amongst themselves.

	AD_0_		AD_1_	
	Feed (S_0_)	Digestate	Feed (S_1_)	Digestate
pH	5.4 ± 0.2 ^a^	7.5 ± 0.1 ^c^	6.5 ± 0.3 ^b^	7.5 ± 0.1 ^c^
EC (mS/cm)	4.5 ± 0.2 ^a^	6.4 ± 0.3 ^b^	6.3 ± 1.5 ^a^	8.3 ± 1.4 ^b^
TS (g/L)	30.9 ± 1.72 ^a^	10.2 ± 1.4 ^b^	33.6 ± 1.93 ^a^	12.8 ± 2.8 ^b^
VS (g/L)	27.2 ± 1.61 ^a^	7.8 ± 1.2 ^b^	28.6 ± 1.8 ^a^	9.3 ± 2.1 ^b^
VS/TS (%)	88 ^a^	75 ^b^	85 ^a^	73 ^b^
VSS (g/L)	-	6.9 ± 1.0 ^b^	-	7.7 ± 1.5 ^b^
TCOD (g/L)	35.8 ± 1.10 ^a^	11.3 ± 1.1 ^b^	38.1 ± 1.6 ^a^	12.5 ± 0.9 ^b^
TKN (g/L)	1.2 ± 0.1 ^a^	1.1 ± 0.1 ^b^	1.1 ± 0.1 ^a^	1.0 ± 0.1 ^b^
N-NH_4_^+^ (g/L)	0.2 ± 0.1 ^a^	0.7 ± 0.1 ^b^	0.3 ± 0.1 ^a^	0.7 ± 0.1 ^b^
C_ORG_ (g/L)	15.8 ± 0.8 ^a^	-	16.6 ± 1.3 ^a^	-
C/N	13 ^a^	-	15 ^a^	-

**Table 4 membranes-13-00371-t004:** Performance and stability parameters. Different letters in the table indicate significantly different results between AD_0_ and AD_1_ (*p* < 0.05) according to Student’s *t*-test. Parameters were not compared amongst themselves.

	AD_0_	AD_1_
GPR (mL/L_reactor_.d)	695 ± 7 ^a^	714 ± 39 ^a^
MPR (mL/L_reactor_.d)	458 ± 4 ^a^	462 ± 30 ^a^
OLR (g SV/L_reactor_.d)	2.09 ± 0.13 ^a^	2.01 ± 0.41 ^a^
CH_4_ (%)	66.5 ± 0.7 ^a^	64.8 ± 1.3 ^a^
SELR (d^−1^)	0.39 ± 0.09 ^a^	0.38 ± 0.03 ^a^

**Table 5 membranes-13-00371-t005:** Feed and digestate characterisation for AcoD trials.

	AcoD	
	Feed Mixture	Digestate
pH	6.7 ± 0.3	7.4 ± 0.1
EC (mS/cm)	6.6 ± 0.9	8.7 ± 0.7
TS (g/L)	29.3 ± 2.3	11.9 ± 1.1
VS (g/L)	23.6 ± 1.7	8.7 ± 0.8
VS/TS (%)	81	73
VSS (g/L)	-	6.8 ± 1.2
TCOD (g/L)	33.3 ± 2.0	12.4 ± 1.2
TKN (g/L)	0.8 ± 0.1	0.7 ± 0.1
N-NH_4_^+^	0.2 ± 0.1	0.4 ± 0.1
C_ORG_	12.6 ± 0.5 14	-
C/N	18	-

**Table 6 membranes-13-00371-t006:** Performance and stability parameters from AcoD trials.

	AcoD
GPR (mL/L_reactor_.d)	571 ± 38
MPR (mL/L_reactor_.d)	373 ± 30
OLR (g SV/L_reactor_.d)	1.32 ± 0.16
CH_4_ (%)	66.3 ± 0.9
SELR (d^−1^)	0.37 ± 0.02

## Data Availability

The data are available from the corresponding author.

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
