# Peer review of "Mango Peel Nanofiltration Concentrates to Enhance Anaerobic Digestion of Slurry from Piglets Fed with Laminaria"

_membranes, 2023, doi:10.3390/membranes13040371_

Round 1

Reviewer 1 Report

The manuscript describes the Nanofiltration concentrates from fruit biowaste to improve the anaerobic digestion of biomass from piglets fed with Laminaria.. I recommend this manuscript for publication in your esteemed journal Membranes, after the following major revisions:

1) In section 2.4 Analytical characterizations, add countries to all material and equipments used in the analysis test like L183 ((Panreac Quimica SAU, Barcelona, Spain)

2) Check the units and the values in table 1. Are you sure that the VS=243…257 g/l

3) The results presented in tables (1-6) should be compared to these obtained in new research papers

4) L300-L302, (The adoption of co-digestion regime led to a more relevant increase in SMP, of 29%, indicating that the Biocon version was more efficient. This fact is probably associated with the more balanced C/N ratio, caused by the co-substrate addition.), please justify your interpretation by adding new references to confirm you idea.

5) Add nomenclature to the manuscript.

6) Revise the title of the manuscript.

7) Add the conditions to all figures of the manuscript.

8) Please highlight the novelty of this study in the manuscript.

9) Recommendations and perspectives are missing in the conclusion section

10) The English language of the manuscript should be re-approved by a native speaker.

Author Response

Dear Reviewer,

Kind regards

Reviewer 2 Report

Comments to the Author:

Title: Nanofiltration concentrates from fruit biowaste to improve the anaerobic digestion of biomass from piglets fed with Laminaria

Overview and general recommendation:

The manuscript deals with an important topic related to the nanofiltration concentrates from fruit biowaste to improve the anaerobic digestion of biomass from piglets fed with Laminaria. The manuscript technically sounds well and shows high novelty. However, it needs moderate linguistic adjustments. In this regard, the needed adjustments are highlighted in “Minor comments” section.

The Abstract part outlines clearly the problematic, aims, methodology and findings of the current study while reporting the main conclusions aroused. The Introduction part is well structured and aiming and underlines appropriately the whole subject under study. However, the aims of the study should be reformulated in a better language besides other linguistic adjustments needed in this part. Also, some statements lacked reliable sources to which some references were recommended. The Materials and methods part is clear, well written, and encloses all the information related to the adopted methodology, and statistical analysis. However, very minor clarifications should be provided regarding the adopted materials. The Results and Discussion part shows a correct statistical representation in some places but overall, it should be much improved. Tables 2, 3 and 5 showed no statistical letters that outline the presence/absence of significant difference. Therefore, they should be added and the whole analysis should be adjusted to outline such significance. Moreover, the discussion of the obtained findings should be improved in almost all this part’s sections. Although some minor linguistic adjustments are needed in the Conclusions part, it summarized appropriately the obtained findings. However, authors shall add a sentence at the end of this section in which they suggest further related research being based on the raised assumptions from the current study.

My comments and queries for authors are detailed below in “Minor comments” section.

1.1.            Major comments:

1-      The manuscript needs moderate linguistic adjustments that can be adjusted appropriately by the authors. The needed adjustments are highlighted in “Minor comments” section.

2-      3. Results and Discussion, 3.1. AD trials: Page 6, line 209, Table 2: Table 2 shows no statistical letters that outline the presence/absence of significant difference. Therefore, they should be added and the whole analysis should be adjusted to outline such significance.

3-      3. Results and Discussion, 3.1. AD trials: Page 6, line 224, Table 3: Same recommendation as in the previous comment.

4-      3. Results and Discussion, 3.2. AcoD trials: Page 7, lined 254, Table 5: Same recommendation as in the previous two comments.

5-      3. Results and Discussion: The discussion of the obtained findings should be improved in almost all this part’s sections.

1.2.            Minor comments:

6-      Abstract: Page 1, lines 12–14: “The need… model”: The sentence is badly written in standard English; accordingly, kindly reformulate it.

7-      Abstract: Page 1, line 14: Kindly replace “is” by “was”.

8-      Abstract: Page 1, line 16: Kindly adjust as follow: “The nanofiltration”.

9-      Abstract: Page 1, lines 18–19: “Slurry… Laminaria”: The sentence is badly written in standard English; accordingly, kindly reformulate it.

10-  1. Introduction: Page 1, line 32: Kindly adjust as follow: “and characterized”.

11-  1. Introduction: Page 1, lines 34–36: “Poor… system”: The sentence is badly written in standard English; accordingly, kindly reformulate it.

12-  1. Introduction: Page 1, lines 36–39: “Moreover… [3,4]”: Same recommendation as in the previous comment.

13-  1. Introduction: Page 1, lines 40–43: “Weaning… stress”: These statements lack reliable sources (references); accordingly, kindly provide them.

14-  1. Introduction: Pages 1–2, lines 43–46: “Thus… [5,6]”: Kindly avoid the first voice form of the sentence and adopt the impersonal form instead.

15-  1. Introduction: Page 2, line 47: Kindly adjust as follow: “the adoption” and “the changes”.

16-  1. Introduction: Page 2, lines 48–49: “Particularly… biowaste”: The sentence is badly written in standard English; accordingly, kindly reformulate it.

17-  1. Introduction: Page 2, lines 54–55: “AD… [12]”: Same recommendation as in the previous comment.

18-  1. Introduction: Page 2, lines 57–58: “To overcome... digestion”: This statement lacks reliable sources (references); accordingly, the following reliable reference is recommended for this statement: “(doi: 10.3390/horticulturae80660479)”.

19-  1. Introduction: Page 2, line 60: Kindly adjust as follow: “crop production”.

20-  1. Introduction: Page 2, lines 59–61: “This biological… performance”: This statement lacks reliable sources (references); accordingly, the following reliable reference is recommended for this statement: “(doi: 10.3390/su141610224)”.

21-  1. Introduction: Page 2, lines 70–72: “In a study… [21,22]”: The sentence is long and cumbersome; accordingly, kindly reformulate in order to make it more concise, clearer and more aiming.

22-  1. Introduction: Page 2, line 74: Kindly adjust as follow: “C/N ratio”.

23-  1. Introduction: Page 2, line 76: Kindly adjust as follow: “the possibility” and “with reduced”.

24-  1. Introduction: Page 2, line 77: Kindly remove “at the same time” and adjust as follow: “they are”.

25-  1. Introduction: Page 2, line 78: Kindly adjust as follow: “and (iii)” and “In a previous work”.

26-  1. Introduction: Page 2, lines 82–83: “The ultrafiltration… permeates”: The sentence is badly written in standard English; accordingly, kindly reformulate it.

27-  1. Introduction: Page 2, lines 86–87: “This work… potential”: Same recommendation as in the previous comment.

28-  2. Materials and Methods, 2.1. Sample collection, pre-treatments and feeding mixtures, 2.1.1. Substrates: Page 3, lines 102–103: “The company… used”: Same recommendation as in the previous two comments.

29-  2. Materials and Methods, 2.1. Sample collection, pre-treatments and feeding mixtures, 2.1.1. Substrates: Page 3, line 104: What was that amount? Kindly mention it.

30-  2. Materials and Methods, 2.1. Sample collection, pre-treatments and feeding mixtures, 2.1.2. Co-substrate: Page 3, lines 117–119: “Since… procedures”: Kindly avoid the first voice form of the sentence and adopt the impersonal form instead.

31-  2. Materials and Methods, 2.1. Sample collection, pre-treatments and feeding mixtures, 2.1.2. Co-substrate: Page 3, lines 119–121: “After… milling”: The sentence is badly written in standard English; accordingly, kindly reformulate it.

32-  2. Materials and Methods, 2.1. Sample collection, pre-treatments and feeding mixtures, 2.1.2. Co-substrate: Page 3, line 123: Kindly add “was reached” after “2.0”.

33-  2. Materials and Methods, 2.1. Sample collection, pre-treatments and feeding mixtures, 2.1.2. Co-substrate: Page 3, line 124: Kindly replace “have” by “had”.

34-  2. Materials and Methods, 2.1. Sample collection, pre-treatments and feeding mixtures, 2.1.2. Co-substrate: Page 3, line 127: Kindly adjust as follow: “and at room temperature”.

35-  2. Materials and Methods, 2.1. Sample collection, pre-treatments and feeding mixtures, 2.1.2. Co-substrate: Page 3, line 131: Kindly adjust as follow: “and kept at room temperature”.

36-  2. Materials and Methods, 2.1. Sample collection, pre-treatments and feeding mixtures, 2.1.2. Co-substrate: Page 4, lines 133–135: “All… used”: These sentences are badly written in standard English; accordingly, kindly reformulate them.

37-  2. Materials and Methods, 2.1. Sample collection, pre-treatments and feeding mixtures, 2.1.3. Co-substrate: Page 4, line 143: Kindly adjust as follow: “ratio” instead of “proportion”.

38-  2. Materials and Methods, 2.2. Experimental design and operational conditions: Page 4, line 149: Kindly replace “has” by “had”.

39-  2. Materials and Methods, 2.2. Experimental design and operational conditions: Page 4, lines 150 and 152: Kindly replace “is” by “was”.

40-  2. Materials and Methods, 2.3. Performance and stability operational parameters: Page 5, line 163: Kindly replace “was” by “were”.

41-  2. Materials and Methods, 2.3. Performance and stability operational parameters: Page 5, line 165: Kindly remove “and” before “specific”.

42-  2. Materials and Methods, 2.4. Analytical characterization: Page 5, line 178: Kindly adjust as follow: “in accordance with”.

43-  3. Results and Discussion, 3.1. AD trials: Page 6, line 201: “Regarding… 5%”: Kindly avoid the first voice form of the sentence and adopt the impersonal form instead.

44-  3. Results and Discussion, 3.1. AD trials: Page 6, line 204: Kindly adjust as follow: “who reported”.

45-  3. Results and Discussion, 3.1. AD trials: Page 6, line 212: Kindly adjust the sentence as follow: “Results in Table 2 highlighted…”

46-  3. Results and Discussion, 3.1. AD trials: Page 6, line 214: Kindly remove “were” and adjust as follow: “(6.5–7.5)”.

47-  3. Results and Discussion, 3.1. AD trials: Page 6, lines 214–216: “Another… [37]”: The sentence is badly written in standard English; accordingly, kindly reformulate it.

48-  3. Results and Discussion, 3.1. AD trials: Page 6, line 217: Kindly adjust as follow: “corresponded”.

49-  3. Results and Discussion, 3.1. AD trials: Page 6, line 220: Kindly adjust as follow: “indicated”.

50-  3. Results and Discussion, 3.1. AD trials: Page 7, line 225: They are not similar!! They can be “comparable” if no statistical difference exists.

51-  3. Results and Discussion, 3.1. AD trials: Page 7, line 226: Kindly adjust as follow: “didn’t have”.

52-  3. Results and Discussion, 3.2. AcoD trials: Page 7, line 232: Kindly adjust as follow: “in a previous”.

53-  3. Results and Discussion, 3.2. AcoD trials: Page 7, line 233: Kindly adjust as follow: “the characterization”.

54-  3. Results and Discussion, 3.2. AcoD trials: Page 7, line 241: Kindly adjust as follow: “concentrations”.

55-  3. Results and Discussion, 3.2. AcoD trials: Page 7, lines 244–246: “This factor… [37]”: The sentence is badly written in standard English; accordingly, kindly reformulate it.

56-  3. Results and Discussion, 3.2. AcoD trials: Page 7, lines 247–249: “The selection… mixture”: Same recommendation as in the previous comment.

57-  3. Results and Discussion, 3.2. AcoD trials: Page 7, lines 250–252: “To corroborate… [36,37]”: Same recommendation as in the previous two comments.

58-  3. Results and Discussion, 3.2. AcoD trials: Page 7, line 253: Kindly adjust as follow: “as well as”.

59-  3. Results and Discussion, 3.2. AcoD trials: Page 8, line 256: Kindly replace “is” by “was”.

60-  3. Results and Discussion, 3.2. AcoD trials: Page 8, line 257: Kindly adjust as follow: “contributed”.

61-  3. Results and Discussion, 3.2. AcoD trials: Page 8, lines 257–261: “As S2… [34]”: The sentence is long and cumbersome; accordingly, kindly reformulate in order to make it more concise, clearer and more aiming.

62-  3. Results and Discussion, 3.2. AcoD trials: Page 8, line 267: Kindly replace “is” by “was”.

63-  3. Results and Discussion, 3.2. AcoD trials: Page 8, lines 268–269: “When comparing… CH4”: The sentence is cumbersome; accordingly, kindly reformulate in order to make it clearer and more aiming.

64-  3. Results and Discussion, 3.2. AcoD trials: Page 8, line 270: Kindly adjust as follow: “the process”.

65-  3. Results and Discussion, 3.2. AcoD trials: Page 9, line 281: Kindly replace “,” by “;”.

66-  3. Results and Discussion, 3.2. AcoD trials: Page 9, line 282: Kindly adjust as follow: “that there was”.

67-  3. Results and Discussion, 3.2. AcoD trials: Page 9, lines 283–284: “This fact… 7.5”: The sentence is badly written in standard English; accordingly, kindly reformulate it.

68-  3. Results and Discussion, 3.2. AcoD trials: Page 9, lines 288 and 290: Kindly replace “are” by “were”.

69-  3. Results and Discussion, 3.2. AcoD trials: Page 9, line 291: Kindly replace “is” by “was”.

70-  3. Results and Discussion, 3.2. AcoD trials: Page 9, line 296: Kindly adjust as follow: “occurred in comparison with”.

71-  3. Results and Discussion, 3.2. AcoD trials: Page 9, line 298: Kindly remove “also for AD with pig slurry”.

72-  4. Conclusions: Page 10, line 303: Kindly adjust the numbering of this part as “4.” instead of “5.”.

73-  4. Conclusions: Page 10, lines 307–309: “Regarding… slurry”: The sentence is long and cumbersome; accordingly, kindly reformulate in order to make it more concise, clearer and more aiming.

74-  4. Conclusions: Page 10, line 311: Kindly adjust as follow: “slight”.

75-  4. Conclusions: Page 10, line 313: Kindly adjust as follow: “increased”.

76-  4. Conclusions: Page 10, line 320: Kindly add a sentence at the end of this part in which you suggest further related research being based on the raised assumptions from the current study.

Author Response

Dear Reviewer,

Kind regards

Round 2

Reviewer 1 Report

Can be published in the current form

Reviewer 2 Report

Comments to the Author:

Title: Nanofiltration concentrates from fruit biowaste to improve the anaerobic digestion of biomass from piglets fed with Laminaria

Overview and general recommendation:

Authors made significant improvements to their manuscript and are well thanked for that. However, some additional improvements to the manuscript’s language and statistical presentation are still needed. But all these adjustments are minor and can be easily dealt by the authors.

My comments and queries for authors are detailed below in “Minor comments” section.

1.1.            Major comments:

1-      No major comments to give.

1.2.            Minor comments:

2-      Abstract: Page 1, line 22: Kindly adjust as follow: “resulting from”.

3-      Abstract: Page 1, line 29: Kindly adjust as follow: “increased by”.

4-      1. Introduction: Page 2, lines 59–61: “Since… employ it”: The sentence is cumbersome; accordingly, kindly reformulate in order to make it clearer and more aiming.

5-      1. Introduction: Pages 1–2, lines 43–46: “Thus… [15,16]”: Kindly avoid the first voice form of the sentence and adopt the impersonal form instead.

6-      Regarding the old comment 18, I am sorry for the typo mistake. The correct doi for the reference to be added is: “(doi: 10.3390/horticulturae8060479)”.

7-      1. Introduction: Page 3, line 119: Kindly adjust as follow: “with reduced”.

8-      1. Introduction: Page 3, line 133: Kindly adjust as follow: “obtained on”.

9-      3. Results and Discussion, 3.1. AD trials: Page 7, lines 267–269: “These results… 13.45”: The sentence is cumbersome; accordingly, kindly reformulate in order to make it clearer and more aiming.

10-  3. Results and Discussion, 3.1. AD trials: Page 7, line 275, Table 2: Kindly adopt lower or upper-case letters but not both.

11-  3. Results and Discussion, 3.1. AD trials: Page 7, line 280: Kindly adjust as follow: “C/N ratio”.

12-  3. Results and Discussion, 3.1. AD trials: Page 7, line 282: Kindly adjust as follow: “between 15 and 30”.

13-  3. Results and Discussion, 3.1. AD trials: Page 8, line 291: Kindly adjust as follow: “The statistical”.

14-  3. Results and Discussion, 3.1. AD trials: Page 8, line 301, Table 3: The comparison is between AD0 and AD1 and not between parameters!! Therefore, kindly place “a” and “b” within the same parameter when there is a significant difference. Same applies to Table 2.

15-  3. Results and Discussion, 3.1. AD trials: Page 8, line 303: Kindly adjust as follow: “didn’t have”.

16-  3. Results and Discussion, 3.2. AcoD trials: Page 9, lines 323–325: “Other… 300 Da”: The sentence is long and cumbersome; accordingly, kindly reformulate in order to make it more concise, clearer and more aiming.

17-  3. Results and Discussion, 3.2. AcoD trials: Page 9, line 331: Kindly replace “is” by “was”.

18-  3. Results and Discussion, 3.2. AcoD trials: Page 9, lines 334–336: “Since… process”: The sentence is cumbersome; accordingly, kindly reformulate in order to make it clearer and more aiming.

19-  3. Results and Discussion, 3.2. AcoD trials: Page 9, line 337: Kindly adjust as follow: “ratio of 80:20 S1:S2”.

20-  3. Results and Discussion, 3.2. AcoD trials: Page 9, line 339: Kindly adjust as follow: “values that are”.

21-  3. Results and Discussion, 3.2. AcoD trials: Page 10, lines 353–357: These sentences are cumbersome; accordingly, kindly reformulate in order to make them clearer and more aiming.

22-  3. Results and Discussion, 3.2. AcoD trials: Page 10, line 368: Kindly adjust as follow: “exhibited”.

23-  3. Results and Discussion, 3.2. AcoD trials: Page 11, line 406: Kindly adjust as follow: “that corresponded”.

24-  4. Conclusions: Page 12, line 427: Kindly remove “In conclusion”.
